# Development and evaluation of multiepitope fusion proteins for serological diagnosis of animal brucellosis

Liping Guo,[1] Qichuan Pei,[1] Shiqi Zhao,[1] Xinru Qi,[1] Yixiao Chen,[1] Peifei Yang,[2] Yangguang Du,[3] Mingjun Sun,[4] Dehui Yin,[1] Tiansong Zhan[1]

**ABSTRACT** Brucellosis, a zoonotic disease caused by *Brucella* spp., leads to severe reproductive issues in livestock and economic losses. Current serological diagnostics using lipopolysaccharide (LPS) antigens often exhibit cross-reactivity, reducing diagnostic specificity. This study aimed to develop multiepitope fusion proteins based on *Brucella* B-cell epitopes using bioinformatics tools and the Immune Epitope Database (IEDB) to enhance diagnostic accuracy. B-cell epitopes from major *Brucella* outer membrane proteins and other antigenic proteins were predicted using bioinformatics tools (BepiPred, ABCpred, and IEDB). Two fusion proteins were designed and produced. The diagnostic performance of the two fusion proteins was evaluated using indirect enzyme-linked immunosorbent assay with 198 small ruminant serum samples and 232 bovine serum samples in comparison with conventional LPS and Rose Bengal antigen. Sensitivity, specificity, and cross-reactivity were evaluated. Both fusion proteins exhibited high sensitivity and specificity. For ruminant samples, Fusion Protein 2 achieved an area under the curve (AUC) of 0.9849, with sensitivity and specificity of 93.90% and 97.26%, respectively. For bovine samples, it showed an AUC of 0.9664, sensitivity of 92.71%, and specificity of 90.44%. Minimal cross-reactivity with other pathogens was observed, indicating high diagnostic specificity. The developed multiepitope fusion proteins demonstrated superior diagnostic performance. These proteins provide a novel tool for rapid and accurate diagnosis of brucellosis, with potential applications in vaccine development and disease control. Future work will focus on optimizing fusion protein design and expanding clinical validation.

**IMPORTANCE** Brucellosis, a zoonotic disease caused by *Brucella* spp., poses a significant threat to livestock industries and human health. Current serological diagnostic methods using LPS antigens often suffer from cross-reactivity, leading to reduced diagnostic specificity. This study addresses this challenge by developing multiepitope fusion proteins based on *Brucella* B-cell epitopes. Using bioinformatics tools and IEDB, we designed and produced two fusion proteins and evaluated their diagnostic performance. The results demonstrated that these fusion proteins exhibited high sensitivity and specificity, with minimal cross-reactivity, offering a more accurate tool for brucellosis diagnosis. This advancement not only enhances the effectiveness of disease surveillance and control but also provides a foundation for potential vaccine development. The successful application of these fusion proteins in serological diagnosis highlights their importance in improving the accuracy and reliability of brucellosis detection, which is crucial for minimizing economic losses and public health risks associated with the disease.

**KEYWORDS** brucellosis, multiepitope fusion proteins, B-cell epitopes, bioinformatics, serological diagnosis

Address correspondence to Mingjun Sun, sunmingjun@cahec.cn, Dehui Yin, yindh16@xzhmu.edu.cn, or Tiansong Zhan, tszhan@xzhmu.edu.cn.

Liping Guo, Qichuan Pei, and Shiqi Zhao contributed equally to this article. Author order was determined both alphabetically and in order of increasing seniority.

The authors declare no conflict of interest.

See the funding table on p. 13.

Brucellosis is a zoonotic infectious disease caused by bacteria of the genus *Brucella*, which affects a wide range of domestic animals, including cattle, sheep, and pigs, and can also be transmitted to humans (1). In domestic animals, the disease can lead to severe consequences, such as abortion, infertility, and testicular inflammation, resulting in significant economic losses for the livestock industry. Serological testing is a crucial tool for diagnosing brucellosis, with commonly used methods including the agglutination test, complement fixation test, and enzyme-linked immunosorbent assay (ELISA) (2, 3). However, the currently utilized lipopolysaccharide (LPS) antigens (Ags) and allozyme antigens exhibit certain cross-reactivity during the detection process, which reduces diagnostic specificity (4, 5). Therefore, the development of more specific and sensitive diagnostic antigens is essential for enhancing the diagnostic accuracy of brucellosis.

In *Brucella*, numerous antigenic proteins play crucial roles in the pathogen's pathogenicity and immune response. For instance, outer membrane proteins (Omps) Omp28 (BP26), Omp16, Omp25, Omp31, Omp2b, DnaK, Elongation Factor Tu (EF-Tu), and Cu-Zn superoxide dismutase (SOD) have been identified as significant in the diagnostic and immunological studies of brucellosis (6–10). BP26, an outer membrane protein, has been extensively utilized in the serological diagnosis of brucellosis and is capable of eliciting a robust immune response (6, 7). Omp16, Omp25, and Omp31 are also vital outer membrane proteins that contribute significantly to brucellosis pathogenesis and demonstrate strong immunogenicity in animal models, making them valuable for diagnostic and vaccine development (8). Omp2b, functioning as a pore protein, is highly immunogenic and is regarded as a promising candidate for diagnostic and vaccine research in brucellosis (9). Additionally, intracellular proteins such as DnaK, EF-Tu, and Cu-Zn SOD play essential roles in the survival and immune evasion of *Brucella* (10–12). Studies have indicated that these proteins can induce a protective immune response in the host, suggesting their potential as diagnostic antigens.

In recent years, bioinformatics technology has been extensively utilized in the prediction of pathogen antigenic epitopes and the design of vaccines. Utilizing bioinformatics tools, B-cell epitopes within pathogen proteins can be predicted rapidly and accurately, providing a theoretical foundation for the development of novel diagnostic antigens. Among the most commonly used methods for predicting linear epitopes are BepiPred, ABCpred, and COBEpro (13–15). BepiPred integrates the hidden Markov model and propensity scale methods to predict linear B-cell epitopes. ABCpred, which relies on a neural network, achieves a prediction accuracy of approximately 65.93%. COBEpro predicts linear B-cell epitopes through a two-step process. Initially, short peptides are predicted using a mechanical model, followed by scoring each amino acid residue. In this study, we selected three prediction tools to enhance accuracy. We combined immunological information parameter prediction with B-cell epitope prediction and compared the B-cell epitopes predicted by the four methods. The overlapping B-cell epitopes were then selected as candidate B-cell epitopes for constructing multiepitope fusion proteins.

In this study, two multiepitope fusion proteins based on B-cell epitopes of *Brucella* were designed and constructed, referencing bioinformatics prediction methods reported in the relevant literature and incorporating data from the Immune Epitope Database (IEDB). The effectiveness of these proteins in the serological diagnosis of brucellosis in animals was subsequently evaluated.

## MATERIALS AND METHODS

### Epitope prediction

The B-cell epitopes of the major Omps and antigenic proteins of *Brucella* were predicted using bioinformatics tools, following the methodology reported in the literature (16). First, the amino acid sequences of the *Brucella* Omps Omp19, Omp16, Omp25, Omp2a, Omp31, Omp2b, BP26, Cu-Zn SOD, DnaK, and EF-Tu were downloaded from the National Center for Biotechnology Information database. These protein sequences were

then analyzed for B-cell epitope prediction using three B-cell linear epitope prediction tools, including ABCpred (https://webs.iiitd.edu.in/raghava/abcpred/index.html, default threshold of 0.5), BCPred (http://ailab-projects2.ist.psu.edu/bcpred/predict.html, default threshold of a specificity of 75%), and BepiPred Linear Epitope Prediction 2.0 (http://tools.iedb.org/bcell/, default threshold of 0.5) (13–15). The predicted B-cell epitopes from all of the tools were compared, and the overlapping B-cell epitopes longer than six amino acids were selected as candidate epitopes. Epitopes predicted by at least two out of the three tools (ABCpred, BCPred, and BepiPred) with overlapping regions were defined as "overlapping epitopes." Only epitopes greater than six amino acids in length were retained.

## Selection of *Brucella* linear B-cell epitopes

The identification of linear B-cell epitopes for *Brucella* was conducted using IEDB (https://www.iedb.org/) with the following basic parameters: epitope type, linear peptide; assay type, B cell; outcome, positive; epitope source, *Brucella*. All other parameters were maintained at their default settings. Epitopes from IEDB were filtered by (i) experimental evidence of B-cell reactivity (monoclonal antibody binding), (ii) positive assay outcomes in *Brucella*-infected hosts, and (iii) exclusion of non-linear or poorly characterized epitopes. The retrieved B-cell epitopes were subsequently optimized for integration. In cases where amino acid sequences overlapped, the epitope was regenerated to ensure that the integrated epitope included the overlapping amino acid sequence.

## Fusion protein design

Based on the predicted B-cell epitopes and the epitope information from IEDB, two fusion proteins were designed, referred to as Fusion Protein 1 and Fusion Protein 2. A flexible connecting peptide, "GGGS," was incorporated between the adjacent epitopes to ensure the spatial stability of the fusion proteins and the immunoreactivity of the epitopes. Epitopes were arranged in the order they appeared in the original protein sequences. Subsequently, the isoelectric point (pI) and molecular weight (MW) of the tandemly linked fusion protein were predicted using ProtParam (http://web.expasy.org/protparam/), available on the ExPASy website. The physicochemical properties of the fusion proteins were thoroughly evaluated. Additionally, the antigenicity of the fusion proteins was predicted and assessed using VaxiJen software (http://www.ddg-pharmfac.net/vaxijen/VaxiJen/VaxiJen.html; threshold: 0.4, default). The three-dimensional (3D) structures of Fusion Protein 1 and Fusion Protein 2 were predicted using the Boltz-1 model. The optimal conformation for each protein was subsequently selected based on the confidence score of the predicted models. Furthermore, based on these selected optimal 3D structural models, the solvent exposure of each residue was quantitatively assessed by calculating the relative solvent accessible surface area (RSASA).

## Preparation of fusion protein

According to the designed amino acid sequence, Beijing Protein Innovation Co., Ltd. was commissioned to perform codon optimization and gene synthesis. Gene synthesis employs the technique of chemical synthesis. The primary approach involves synthesizing short-chain DNA on a solid-phase support via chemical reactions. Subsequently, these short-chain DNA fragments are progressively assembled into long-chain DNA through the polymerase chain reaction. Finally, the synthesized sequence is ligated to the appropriate vector using restriction enzyme digestion. The synthesized gene was then cloned into the prokaryotic expression vector pET30a to construct the recombinant expression plasmid. A C-terminal 6× His tag was appended to facilitate purification. This plasmid was transformed into *Escherichia coli* BL21 competent cells, and positive clones were selected for induced expression. When the optical density at 600 nm of the bacterial culture reached 0.6–0.8, 0.5 mM isopropyl β-D-1-thiogalactopyranoside was added to induce expression for 4 h. Following induction, the bacterial pellet was

collected and resuspended in a 10 mM Tris-HCl (pH 8.0) solution before being subjected to ultrasonic disruption. The lysed bacterial solution was centrifuged, and the supernatant underwent SDS-PAGE electrophoretic analysis to detect the expression of the fusion protein.

In the context of inclusion body protein purification, it is imperative to conduct renaturation prior to the purification process. This procedure comprises several critical steps: 1 mL of the sample is taken and added to 20 mL of dialysis buffer containing 1% glycine, 0.1% SDS, 5% glycerol, and 10 mM Tris-HCl (pH 8.0) for sample dialysis. The sample is dialyzed using urea concentrations of 6, 4, and 2 M, facilitating protein renaturation at 4°C. Each urea concentration gradient requires 3 h of dialysis, with the final step involving overnight dialysis. Subsequently, the sample undergoes two dialysis steps in 500 mL of buffer containing 1% glycine and 10 mM Tris-HCl (pH 8.0), each lasting 3 h. Ultimately, the solution is centrifuged at $10,000 \times g$ for 10 minutes, and the supernatant is collected.

The supernatant containing the fusion protein was purified using a Ni-NTA affinity chromatography column, and the target protein peaks were collected by elution with buffers containing varying concentrations of imidazole. The purity of the purified proteins was evaluated through SDS-PAGE electrophoresis and subsequently analyzed using Image Lab software (Bio-Rad, USA). Additionally, the concentration of the proteins was determined using a BCA protein quantification kit. The purified protein underwent endotoxin removal and quantification using the Protein Endotoxin Removal Kit (Beyotime, C0268S) and the Endotoxin Detection Kit (limulus reagent dynamic turbidimetric method) (Beyotime, C0271S). Endotoxin removal and detection were performed in accordance with the kit instructions (Supporting Information 1).

## Serum sample collection and testing

A total of 198 serum samples were collected from small ruminants, comprising 82 positive and 116 negative samples, alongside 232 serum samples from bovines, which included 96 positive and 136 negative samples. All positive samples were confirmed using the Rose Bengal plate agglutination test (RBPT) and the test tube agglutination test (SAT). Negative samples were sourced from regions where brucellosis is not endemic. The samples were provided by the Chinese Center for Animal Hygiene and Epidemiology.

Serum samples were tested using the indirect enzyme-linked immunosorbent assay (iELISA) method, employing purified Fusion Proteins 1 and 2, LPS, and Rose Bengal Ag as test antigens. The procedure was as follows: antigens were diluted to a concentration of 1.0 µg/mL in coating buffer solution (carbonate buffer solution, pH 9.6) and subsequently added to a 96-well ELISA plate (Corning, USA) at a volume of 100 µL per well. The plate was then incubated at 4°C overnight. After washing with phosphate-buffered saline with Tween 20 (PBST) three times, a blocking solution consisting of 5% skimmed milk powder (prepared in phosphate-buffered saline) was added and incubated at 37°C for 2 h. Following another three washes with PBST, serum samples diluted at 1:400 were added and incubated at 37°C for 1 h. After three additional washes with PBST, a 1:10,000 dilution of horseradish peroxidase-conjugated protein G (Thermo Fisher, USA) was added, and the reaction was carried out at room temperature for 30 minutes. After that, the samples were washed three more times with PBST, and TMB (TCI, Japan) color development solution was added. Then the reaction was incubated in the dark for 10 minutes, after which it was terminated by the addition of 2 M $H_2SO_4$. The $OD_{450}$ was measured via a microplate reader (VersaMax Microplate Reader, Molecular Devices [MD], USA), and each sample was tested in triplicate.

## Evaluation of cross-reactivity

To assess the specificity of the fusion proteins, six common pathogens, including *Escherichia coli* O157:H7, *Yersinia enterocolitica* O9, *Salmonella enterica* (A-I), *Vibrio parahaemolyticus*, *Vibrio cholerae*, and *Listeria monocytogenes*-infected rabbit serum

samples (purchased from Tianjin Biochip Corporation, Tianjin, China) were evaluated for cross-reactivity by iELISA. The procedure followed the same protocol as outlined in Serum Sample Collection and Testing. The $OD_{450}$ was measured, and the ratio of the positive serum $OD_{450}$ value (sample [S]) to the negative serum $OD_{450}$ value (negative [N] control) was calculated. A positive result was defined as a signal-to-noise ratio (S/N) of ≥2.1, while a negative result was indicated by a ratio (S/N) of <2.1. These results determine the method's ability to accurately assess outcomes and evaluate its analytical specificity.

## Statistical methods

Data were analyzed using GraphPad Prism software. The diagnostic performance of each antigen was assessed through receiver operating characteristic curve analysis, and the area under the curve (AUC), sensitivity, specificity, positive predictive value (PPV), and negative predictive value (NPV) were calculated. Additionally, the data between different groups were statistically analyzed using the Student's $t$-test (unpaired $t$-test), with a $P$ value of less than 0.05 indicating that the difference was statistically significant.

## RESULTS

### Design of fusion proteins

A total of 42 B-cell epitopes were predicted in Omp19, Omp16, Omp25, Omp2a, Omp31, Omp2b, BP26, Cu-Zn SOD, DnaK, and EF-Tu in individuals, constituting Fusion Protein 1 (see Table 1). The predicted MW was approximately 68.6 kDa, with a pI of 4.64. The overall prediction for the protective antigen was calculated to be 1.5442, indicating a probable antigen.

In contrast, a total of 23 epitopes from the Omp31, Omp2b, BP26, Cu-Zn SOD, DnaK, and EF-Tu proteins were identified in IEDB. Subsequent optimization and integration processes led to the identification of 11 epitopes used to construct Fusion Protein 2 (refer to Table 2 and Supporting Information 2). The MW of the constructed fusion protein was predicted to be approximately 35.1 kDa, and its theoretical pI was determined to be 4.57. The overall prediction for the protective antigen was calculated to be 0.9559, indicating a probable antigen.

The predicted 3D cartoon models of Fusion Protein 1 and Fusion Protein 2 provide a visual representation of their overall folding characteristics (Fig. 1A and B). To further quantify their surface exposure properties, the RSASA was analyzed for both fusion proteins. The results indicated that both proteins possess ample surface-exposed regions (RSASA >0.5), suggesting a high probability that multiple antigenic epitopes within these fusion proteins are located in these solvent-accessible areas, thereby facilitating effective interaction with target antibodies in serum. Concurrently, a subset of residues exhibited partial exposure (0.2 ≤ RSASA ≤ o.5), forming transitional zones between completely buried and fully exposed regions; these residues may play roles in modulating protein flexibility, conformational changes, or serving as secondary contact points for interactions. Additionally, both fusion proteins formed stable internal core structures (RSASA <0.2), which are crucial for maintaining the correct protein conformation, ensuring the proper spatial presentation of epitopes, and for the long-term stability of the diagnostic reagent.

### Expression and purification of fusion proteins

SDS-PAGE results indicated that Fusion Protein 1 (inclusion body) and Fusion Protein 2 (soluble form) were successfully produced through prokaryotic expression. The purified fusion proteins exhibited over 90% purity, which is sufficient for subsequent experiments (see Fig. 2 and Supporting Information 3).

Following testing, the endotoxin levels of purified Fusion Protein 1 and Fusion Protein 2 were determined to be 24.55 and 17.18 EU/mL, respectively. After endotoxin removal,

**TABLE 1** The predicted B-cell epitopes of Fusion Protein 1 by bioinformatics

| Protein source (accession no.) | Epitope | Start-end position |
|---|---|---|
| BP26 (AEF59020) | NQMTTQPARIAV | 31–42 |
| | QPIYVYPDDKNNLKEPTIT | 104–122 |
| | RPPMPMPIARG | 206–216 |
| | APDNSVPIAAGENSYNVSVVNVVFE | 225–248 |
| Omp19 (WCB88146.1) | NLDNVSPPPPPAPVNAVPASTV | 28–49 |
| | KGNLDSPTQFPNAPSTDMSAQSGTQV | 51–76 |
| | AFAPDLTPG | 82–90 |
| | QTKYGQGY | 111–118 |
| | QGRFDGQTTGG | 160–170 |
| Omp16 (AEF59023) | NLPNNAGDLGL | 30–40 |
| Omp25 (AEF59022) | QPPVPAPVEV | 30–39 |
| | TSTVGSIKP | 63–71 |
| | NGLDDES | 147–153 |
| Omp2a (AKN23427.1) | NNSRHDGQYGDFSDDRDVADGGVS | 116–139 |
| | GGEDVDND | 208–215 |
| | SSAATPNQNYGQWG | 274–287 |
| | TKFGGEWKDTV | 341–351 |
| Omp2b (AGO95097.1) | KGGDDVYSGTDRNGWD | 81–96 |
| | NNSGVDGKYGNETSSG | 127–143 |
| | NDGGYTGTTNYHI | 214–226 |
| | PDQNYGQWGG | 287–296 |
| | VSYIKFGGEWKNTVAEDN | 346–363 |
| Omp31 (ACS50328.1) | VSEPSAPTAAP | 24–34 |
| | FDKEDNEQVS | 63–72 |
| | QAGYNWQLDNGVVLGA | 87–102 |
| | GDDASALHMW | 168–177 |
| Cu-Zn SOD (ADL60381.1) | EKLTPGY | 57–63 |
| | SCAPGEKDGKIVP | 73–85 |
| | NTHHHLGPEGDGHMG | 97–111 |
| | PHLKKL | 131–136 |
| | MVHVGGDNYSDKPEPL | 145–160 |
| DnaK (AIJ89704.1) | RRYDDPMVTKDKDLVP | 75–90 |
| | GKEPHKGVN | 353–361 |
| | TRLIERNTTIPTKKS | 404–418 |
| | IPPAPRGV | 457–464 |
| | DKGTGKEHQ | 485–493 |
| | DAEANAEADKKRRESVEAKN | 513–532 |
| | EGAGAEGGEQASSSKDDVVD | 605–624 |
| EF-Tu (AAL51936.1) | SKYEFPGD | 154–161 |
| | IPTPERPI | 197–204 |
| | DEGGRHTPFFT | 312–322 |
| | GTEMVMP | 344–350 |

the endotoxin levels of both proteins were reduced to 0.132 and 0.084 EU/mL, respectively. The standard curve is shown in Supporting Information 1.

## iELISA results

### Results of ruminant serum samples

For ruminant serum samples, the AUC value for Fusion Protein 1 was 0.9632 (95% confidence interval [CI]: 0.9395–0.9869). The sensitivity was 81.71% (95% CI: 0.7163–0.8938), and the specificity was 97.95% (95% CI: 0.9411–0.9957). The PPV was 95.71%,

**TABLE 2** The entire optimized B-cell epitopes of Fusion Protein 2 from IEDB

| Protein (source organism) | Epitope | Start-end position |
|---|---|---|
| BP26 (*Brucella melitensis*) | TMLAAAPDNSVPIAAGENSYNVSVNVVFEIK | 220–250 |
| | KKAGIEDRDLQTGGINIQPIYVYPDDKNNLKEPTITGY | 87–117 |
| DnaK (*B. melitensis*) | SSKDDVVDADYEEIDDNKKSS | 617–637 |
| EF-Tu (*B. melitensis*) | QTREHIL | 110–116 |
| Omp2b (*B. melitensis*) | VIEEWAAKVRGDVNITDQFSVWLQGAYSSAATPDQNYGQWG | 255–295 |
| Omp31 (*B. melitensis/Brucella ovis*) | SWTGGYIGINAGYAGGKFKHPFSSFDKEDNEQVSGSLDVTAGGFV | 39–83 |
| | QAGYNWQLDNGVVLGA | 87–102 |
| | MVYGTGGLAYGKVKSAFNLGDDAPALHTWSDKTKAGWTLGAGAE | 149–192 |
| | EYLYTDLGKRNLVDVDNSFL | 204–223 |
| Cu-Zn SOD (*Brucella abortus*) | APGEKDGKIVPA | 75–86 |
| | LAEIKQRSLMVHVGGDNYSDKPEPLGG | 136–162 |

and the NPV was 88.28%. Fusion Protein 2 exhibited an AUC value of 0.9849 (95% CI: 0.9714–0.9983), with a sensitivity of 93.90% (95% CI: 0.8634–0.9799) and a specificity of 97.26% (95% CI: 0.9313–0.9925). The PPV was 95.06%, and the NPV was 95.73%. In comparison, LPS had an AUC of 0.9945 (95% CI: 0.9888–1.000), with a sensitivity of 97.56% (95% CI: 0.9147–0.9970) and a specificity of 96.55% (95% CI: 0.9141–0.9905). The PPV was 95.24%, and the NPV was 98.25%. The AUC value for the Rose Bengale Ag was 0.9922 (95% CI: 0.9850–0.9995), with a sensitivity of 96.34% (95% CI: 0.8968–0.9924) and a specificity of 94.83% (95% CI: 0.8908–0.9808). The PPV was 92.94%, and the NPV was 97.35% (see Fig. 3; Table 3).

### Detection results of bovine serum samples

For bovine serum samples, the AUC value for Fusion Protein 1 was 0.8954 (95% CI: 0.8562–0.9345). The sensitivity was 80.21% (95% CI: 0.8689–0.9767), and the specificity was 0.8382% (95% CI: 0.7654–0.8958), with a PPV of 77.78% and an NPV of 85.71%. The AUC value for Fusion Protein 2 was 0.9664 (95% CI: 0.9445–0.9883), with a sensitivity of 92.71% (95% CI: 0.8555–0.9702), specificity of 90.44% (95% CI: 0.8421–0.9481), PPV of 87.25%, and NPV of 94.61%. The AUC value for LPS was 0.9796 (95% CI: 0.9657–0.9935), with a sensitivity of 90.63% (95% CI: 0.8295–0.9562), specificity of 95.59% (95% CI: 0.9064–0.9836), PPV of 93.55%, and NPV of 93.53%. The Rose Bengale Ag exhibited an AUC value of 0.9765 (95% CI: 0.9607–0.9923), with a sensitivity of 97.91% (95% CI: 0.9268–0.9975), specificity of 88.24% (95% CI: 0.8160–0.9312), PPV of 85.45%, and NPV of 98.36% (see Fig. 4; Table 3).

### The results of the cross-reactivity assessment

The results indicated that neither Fusion Protein 1 nor Fusion Protein 2 exhibited significant cross-reactivity with these pathogens, demonstrating high specificity (see Table 4 and Supporting Information 4).

### DISCUSSION

iELISA is recommended by the OIE (World Organization for Animal Health) for various purposes, including assessing population freedom from infection, determining individual animal freedom from infection, contributing to eradication policies, confirming suspected or clinical cases, and monitoring herd or flock prevalence of infection in animals (https://www.woah.org/en/disease/brucellosis/). However, traditional iELISA used for diagnosing brucellosis typically employs LPS as the antigen. LPS is commonly found in gram-negative bacteria and has been shown to cross-react with *Escherichia coli* O157:H7 and *Yersinia enterocolitica* O:9 when used as a diagnostic antigen (4, 5). Therefore, identifying more specific diagnostic antigens could potentially enhance the effectiveness of iELISA.

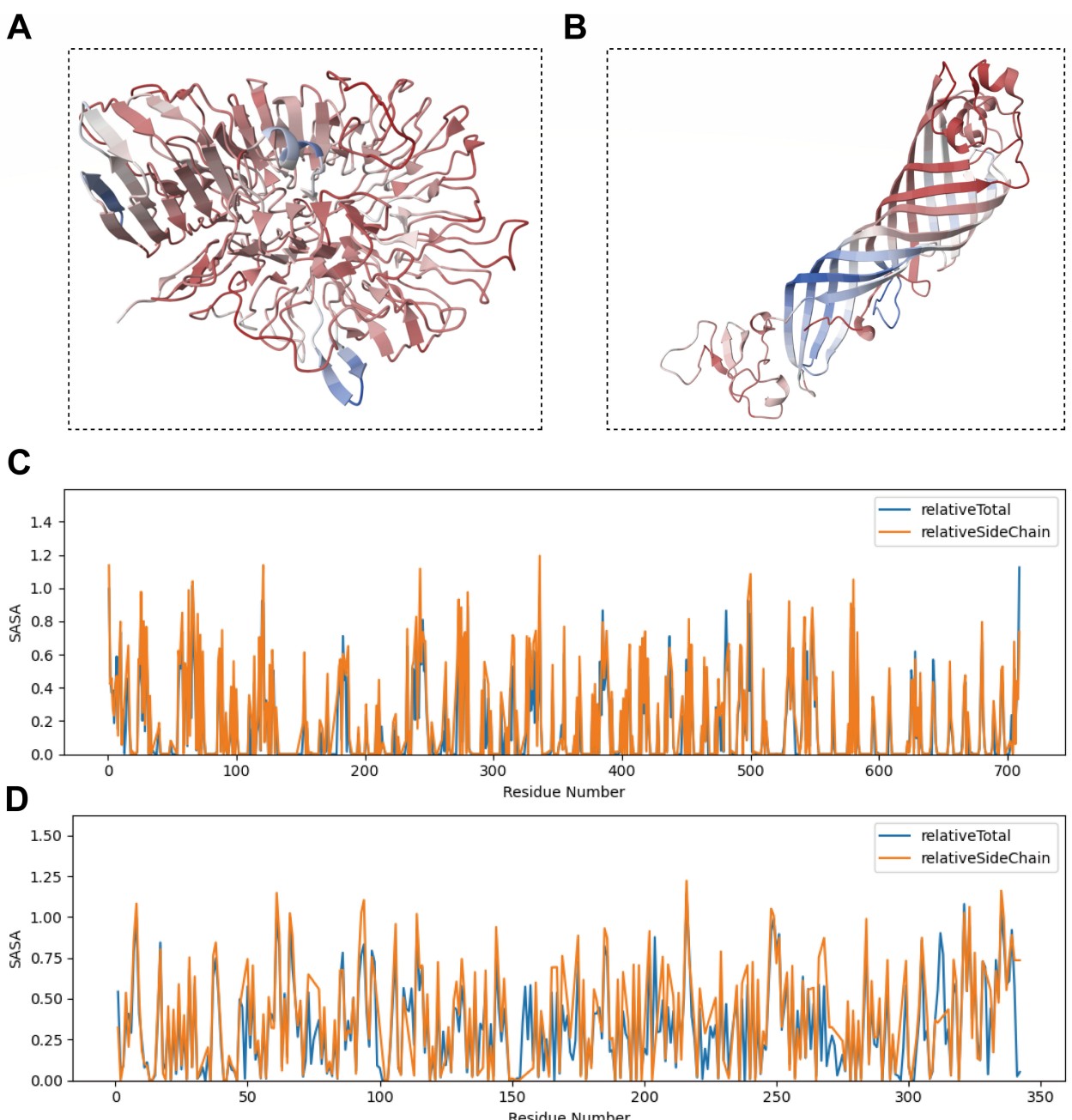

**FIG 1** Predicted 3D structures and relative solvent accessible surface area (RSASA) analysis of the two fusion proteins. (A) Predicted 3D structure (cartoon model) of Fusion Protein 1. (B) Predicted 3D structure (cartoon model) of Fusion Protein 2. (C) Line plot showing the RSASA versus residue number for Fusion Protein 1. (D) Line plot showing the RSASA versus residue number for Fusion Protein 2. (C and D) The blue curve represents the total RSASA (relativeTotal), and the orange curve represents the side-chain RSASA (relativeSideChain) for each amino acid residue. The RSASA value reflects the degree of solvent exposure for each residue, where RSASA >0.5 indicates a solvent-exposed residue; RSASA <0.2 indicates a buried residue; and 0.2 ≤ RSASA ≤0.5 indicates a partially exposed residue.

Omp16, Omp19, Omp25, Omp2a, Omp31, Omp2b, BP26, Cu-Zn SOD, DnaK, and EF-Tu are antigenic proteins with significant immunogenic properties in *Brucella*. Omp16, Omp19, Omp25, Omp31, and BP26 are key outer membrane proteins of *Brucella* that exhibit strong immunogenicity and can elicit a robust immune response in the host. These proteins have been shown to be valuable for the diagnosis of brucellosis in serological assays (6–9, 17, 18). Omp2b, which functions as a pore protein, demonstrates a high level of immunogenicity and is considered a promising candidate for brucellosis

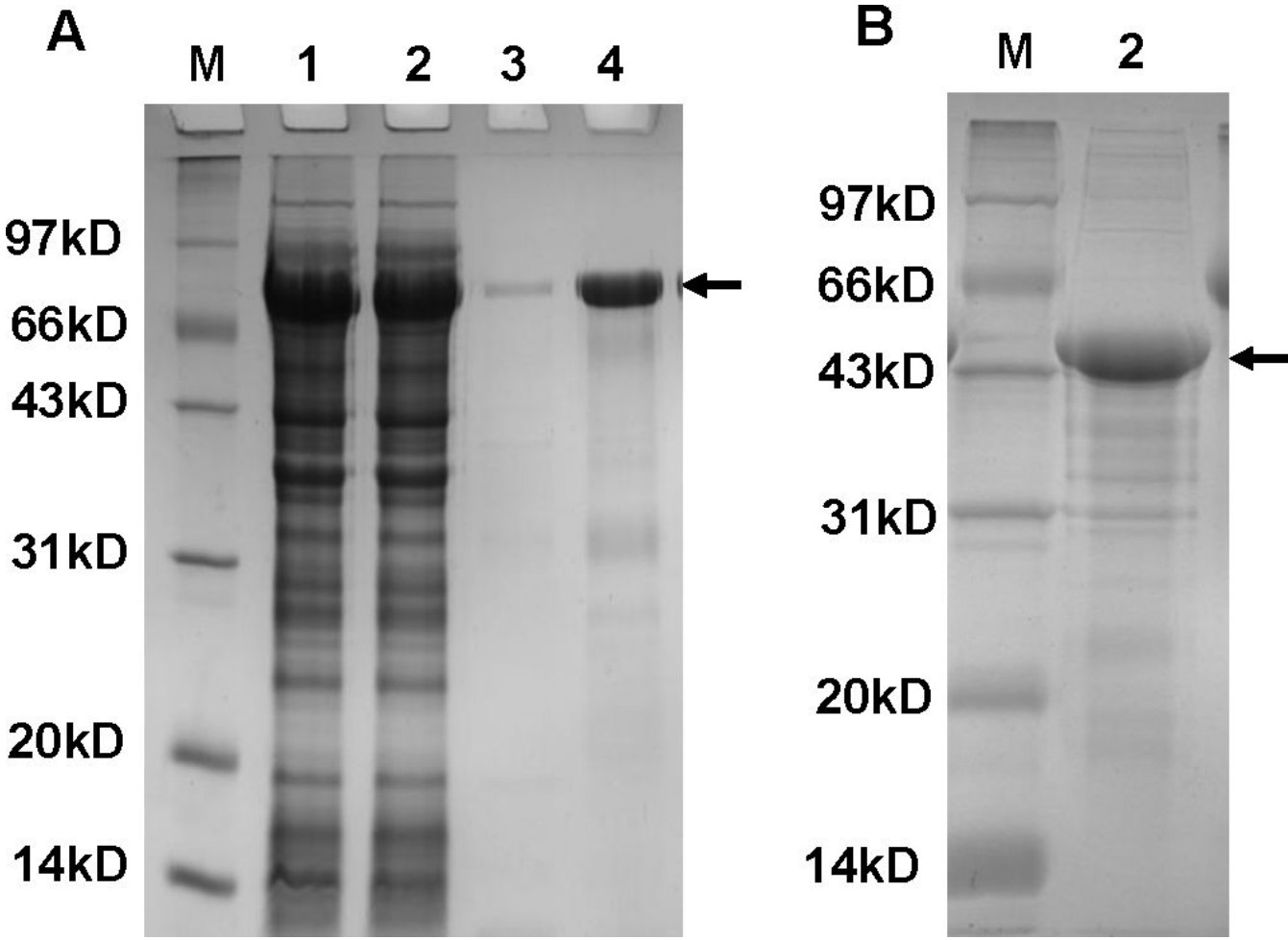

**FIG 2** Twelve percent SDS-PAGE results. (A) Fusion Protein 1: M; lanes 1 and 2, unpurified protein stock solutions; lane 3, purified fusion protein (10-fold dilution); lane 4, purified fusion protein. (B) Fusion Protein 2: M; lane 2, purified fusion protein. M, marker.

diagnosis and vaccine development (9, 19). Cu-Zn SOD is an intracellular protein that plays a crucial role in the survival and immune evasion of *Brucella*, capable of inducing a protective immune response in the host (11, 20). DnaK and EF-Tu are additional intracellular proteins that are essential for the stress response and protein synthesis in *Brucella*, and they also have the potential to induce a protective immune response (12, 13, 21). Furthermore, these proteins may serve as diagnostic antigens due to their ability to elicit an immune response in the host.

The predicted 42 epitopes of these proteins were integrated into fusion proteins based on predictions made by bioinformatics tools. The resulting fusion proteins exhibited high sensitivity and specificity in the serological diagnosis of brucellosis in animals. In this study, we also selected B-cell epitopes cataloged in IEDB for the tandem construction of multiepitope fusion proteins, which were subsequently validated experimentally. This approach addressed the limitations associated with inaccurate bioinformatics predictions. A total of 23 epitopes derived from six antigenic proteins within the selected IEDB collection were validated using monoclonal antibodies (7, 10, 12, 22–30). These previously validated epitopes served as a foundation for the design of our multiepitope fusion proteins. Based on the 23 epitopes and subsequent integration and optimization efforts, we developed a fusion protein that incorporates 11 linear B-cell epitopes. The results of this study confirmed that the protein demonstrated effective performance in diagnosing brucellosis, with detection results surpassing those obtained

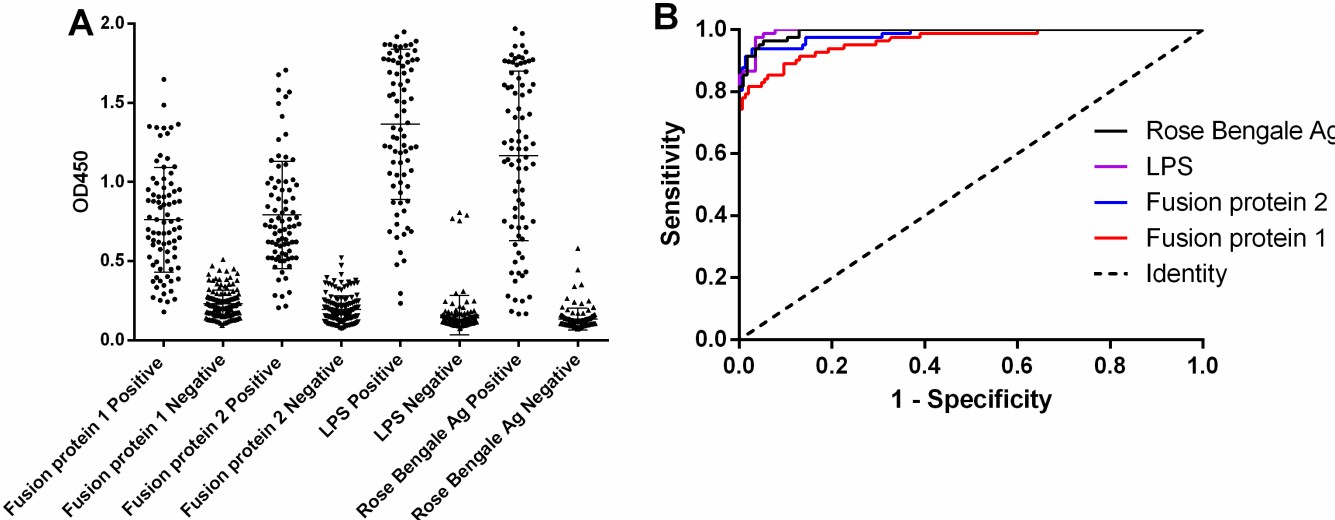

**FIG 3** iELISA analysis of small ruminant serum samples. (A) Dot plot of small ruminant serum samples. (B) Receiver operating characteristic analysis of the small ruminant sera.

using LPS. However, it should be acknowledged that the diagnostic reliability of Fusion Protein 1 may be affected by its lack of experimental validation of the predicted epitopes. Although the epitopes were selected based on bioinformatics predictions, this approach may introduce some uncertainty and potential inaccuracies in the predicted epitopes. In contrast, Fusion Protein 2, supported by monoclonal antibody mapping, has a more solid experimental basis. Therefore, the diagnostic performance of Fusion Protein 1 should be interpreted with caution. In future studies, efforts should be made to further validate the predicted epitopes of Fusion Protein 1 through experimental methods, such as monoclonal antibody binding assays or peptide microarrays. This will help to improve the reliability and specificity of Fusion Protein 1 and provide more robust evidence for its diagnostic potential in brucellosis.

In this study, two multiepitope fusion proteins based on *Brucella* B-cell epitopes were successfully constructed and utilized for the serological diagnosis of brucellosis in animals. The results demonstrated that both fusion proteins exhibited high sensitivity and specificity in diagnosing small ruminants and bovine brucellosis, with reduced cross-reactivity compared to conventional LPS antigen and Rose Bengal Ag. This indicates that the fusion proteins can more accurately identify serum samples from brucellosis-infected animals, thereby minimizing the risk of misdiagnosis.

The diagnostic performance of Fusion Protein 2 was superior to that of Fusion Protein 1 in detecting ruminant serum samples, exhibiting higher AUC values, sensitivity, and

**TABLE 3** Evaluation of iELISA results of the fusion proteins[c]

| Antigen | AUC | Cut-off value | Sensitivity (95% CI) | Specificity (95% CI) | Positive | | Negative | | Accuracy | PPV (%) | NPV |
| --- | --- | --- | --- | --- | --- | --- | --- | --- | --- | --- | --- |
| | | | | | TP | FN | TN | FP | (%) | | (%) |
| Fusion Protein 1[a] | 0.9632 (0.9395–0.9869) | >0.4449 | 0.8171 (0.7163–0.8938) | 0.9795 (0.9411–0.9957) | 67 | 15 | 113 | 3 | 90.91 | 95.71 | 88.28 |
| Fusion Protein 2[a] | 0.9849 (0.9714–0.9983) | >0.3789 | 0.9390 (0.8634–0.9799) | 0.9726 (0.9313–0.9925) | 77 | 5 | 112 | 4 | 95.45 | 95.06 | 95.73 |
| Rose Bengal Ag[a] | 0.9922 (0.9850–0.9995) | >0.2445 | 0.9634 (0.8968–0.9924) | 0.9483 (0.8908–0.9808) | 79 | 3 | 110 | 6 | 95.45 | 92.94 | 97.35 |
| LPS[a] | 0.9945 (0.9888–1.000) | >0.3930 | 0.9756 (0.9147–0.9970) | 0.9655 (0.9141–0.9905) | 80 | 2 | 112 | 4 | 96.97 | 95.24 | 98.25 |
| Fusion Protein 1[b] | 0.8954 (0.8562–0.9345) | >0.2251 | 0.8021 (0.8689–0.9767) | 0.8382 (0.7654–0.8958) | 77 | 19 | 114 | 22 | 82.33 | 77.78 | 85.71 |
| Fusion Protein 2[b] | 0.9664 (0.9445–0.9883) | >0.2755 | 0.9271 (0.8555–0.9702) | 0.9044 (0.8421–0.9481) | 89 | 7 | 123 | 13 | 91.38 | 87.25 | 94.61 |
| Rose Bengale Ag[b] | 0.9765 (0.9607–0.9923) | >0.1855 | 0.9791 (0.9268–0.9975) | 0.8824 (0.8160–0.9312) | 94 | 2 | 120 | 16 | 92.24 | 85.45 | 98.36 |
| LPS[b] | 0.9796 (0.9657–0.9935) | >0.3297 | 0.9063 (0.8295–0.9562) | 0.9559 (0.9064–0.9836) | 87 | 9 | 130 | 6 | 93.53 | 93.55 | 93.53 |

[a]Ruminants.
[b]Bovine.
[c]Ag, antigen; FN, false negative; FP, false positive; TN, true negative; TP, true positive.

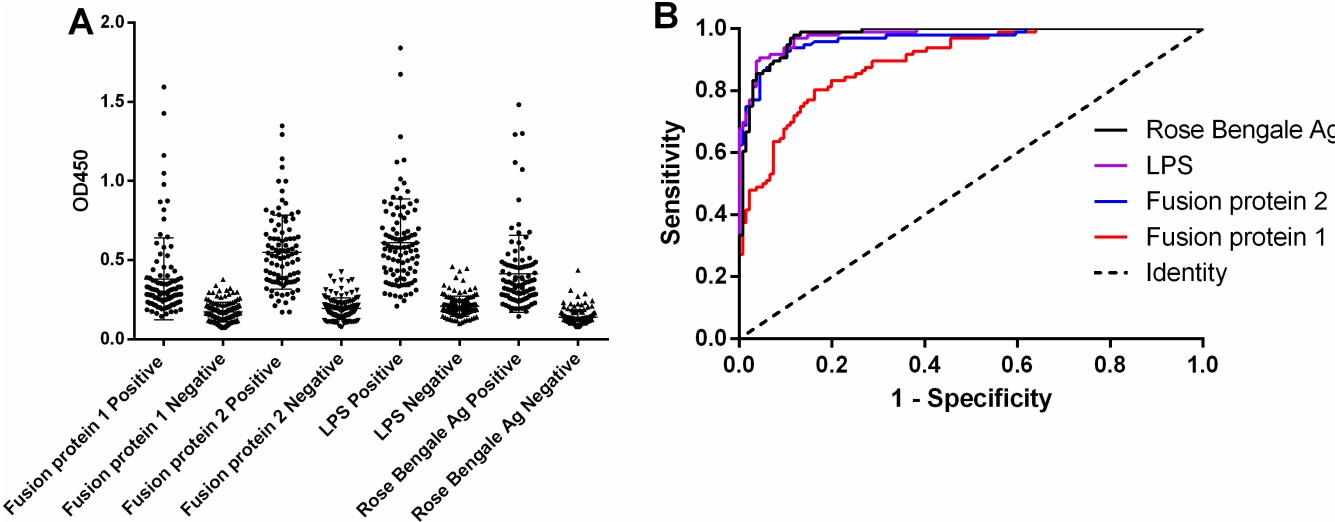

**FIG 4** iELISA analysis of bovine serum samples. (A) Dot plot of bovine serum samples. (B) Receiver operating characteristic analysis of bovine sera.

specificity compared to Fusion Protein 1, and showing comparable results to LPS and Rose B antigen. This superiority may be attributed to the more rigorous screening and validation of the epitopes contained in Fusion Protein 2 within IEDB. Although Fusion Protein 1 offers more comprehensive coverage of *Brucella* antigenic proteins, the B-cell epitopes predicted by bioinformatics techniques are subject to uncertainty and may lack accuracy. In the detection of bovine serum samples, the diagnostic performance of the two fusion proteins was more similar; however, the specificity of Fusion Protein 2 was slightly higher than that of Fusion Protein 1, indicating an advantage in reducing false-positive results. It is important to note that the specificity of the prepared Fusion Protein 2 antigens was somewhat inferior to that of LPS in certain serological assays. This discrepancy may be due to inadequate purification of the two proteins and the presence of impurities affecting the assay results.

In addition, the results of the cross-reactivity test demonstrated that both fusion proteins exhibited high specificity and did not produce significant cross-reactivity with serum samples from non-*Brucella*-infected animals. This finding aligns with the literature (8): LPS and Rose Bengal Ag are prone to cross-reactivity with other pathogens, which indicates that fusion proteins based on *Brucella* outer membrane protein epitopes possess superior specificity and can effectively differentiate *Brucella* infections from those caused by other pathogens. This characteristic is crucial for enhancing the diagnostic accuracy of brucellosis, particularly in the prevention and control of animal diseases, thereby mitigating unnecessary economic losses resulting from misdiagnosis.

However, this study has several limitations. First, the sample size was relatively small, particularly concerning the bovine serum samples, and the number of negative samples

**TABLE 4** Cross-reactivity test results of the fusion proteins

| Pathogens | Fusion Protein 1 | | Fusion Protein 2 | |
|---|---|---|---|---|
| | OD$_{450}$ | S/N | OD$_{450}$ | S/N |
| *E. coli* O157:H7 | 0.0806 | 0.8611 | 0.0884 | 0.6154 |
| *Legionella pneumophila* | 0.1211 | 1.2934 | 0.1797 | 1.2503 |
| *Salmonella* | 0.0922 | 0.9847 | 0.1672 | 1.1635 |
| *Yersinia enterocolitica* O9 | 0.0775 | 0.8280 | 0.0770 | 0.5356 |
| *Vibrio parahaemolyticus* | 0.1308 | 1.3974 | 0.2793 | 1.9434 |
| *Listeria monocytogenes* | 0.0842 | 0.8992 | 0.1058 | 0.7360 |
| *Vibrio cholerae* | 0.1055 | 1.1268 | 0.0917 | 0.6381 |
| Negative control | 0.0936 | 1.0 | 0.1437 | 1.0 |

was limited, which may have impacted the reliability of the statistical results. Future research should aim to increase the sample size to further validate the diagnostic performance of the fusion protein across different regions and animal species. The positive serum samples were confirmed using the RBPT and the SAT, which are well-established serological methods. However, it is important to note that these methods do not provide molecular confirmation of *Brucella* infection. Future studies should consider incorporating molecular methods such as PCR to further corroborate the diagnostic results. Additionally, the collection of whole blood or other sample types could provide additional insights and allow for a more comprehensive evaluation of the diagnostic performance of the fusion proteins against a range of sample types and in comparison with other diagnostic techniques. Second, this study focused solely on evaluating the serological diagnostic efficacy of the fusion proteins and did not investigate their potential for vaccine development or other applications. Subsequent studies could explore the use of fusion proteins as candidate vaccine antigens for assessing immune efficacy or developing more effective brucellosis diagnostic protocols by integrating them with other diagnostic techniques, such as molecular biology detection methods. Lastly, epitopes were arranged in accordance with their sequence in the original proteins, and a linker was employed to facilitate proper folding. While this approach successfully enabled the construction and expression of fusion proteins, it is probable that it did not fully optimize antigenicity. Future research should focus on optimizing the order of epitopes by utilizing immunogenicity scores and incorporating linkers with specified lengths and sequences to enhance spatial separation and stability, among other factors.

## Conclusions

In this study, two multiepitope fusion proteins were successfully constructed by predicting and screening *Brucella* B-cell epitopes using bioinformatics tools and IEDB. These proteins were applied to the serological diagnosis of brucellosis in animals. The results demonstrated that the two fusion proteins exhibited high sensitivity and specificity in diagnosing ruminant and bovine brucellosis, with lower cross-reactivity compared to conventional antigens. This finding provides a novel technical tool for the rapid and accurate diagnosis of brucellosis, which is anticipated to play a significant role in the prevention and control of animal diseases. Future studies will further optimize the design and preparation process of the fusion proteins, expand the sample size for clinical validation, and explore their potential applications in areas such as vaccine development.

### ACKNOWLEDGMENTS

We thank the China Animal Health and Epidemiology Center for the gift of lipopolysaccharides, Brucella strains, and sera.

This work was supported by Xuzhou Science and Technology Bureau (grant number KC23306), the Medical Research Program of Jiangsu Commission of Health (grant number Z2023080), and QingLan Project of Jiangsu Province (2024). The funders had no role in study design, data collection and analysis, decision to publish, or preparation of the manuscript.

### AUTHOR AFFILIATIONS

[1]Jiangsu Engineering Research Center of Biological Data Mining and Healthcare Transformation, Xuzhou Medical University, Xuzhou, China
[2]Huai'an Center for Disease Control and Prevention, Huai'an, China
[3]Xuzhou Center for Disease Control and Prevention, Xuzhou, China
[4]Laboratory of Zoonoses, China Animal Health and Epidemiology Center, Qingdao, China

### AUTHOR ORCIDs

Mingjun Sun http://orcid.org/0000-0002-9194-7333

Dehui Yin ⬤ http://orcid.org/0000-0002-7164-9320
Tiansong Zhan ⬤ http://orcid.org/0000-0001-9793-7139

## FUNDING

| Funder | Grant(s) | Author(s) |
|---|---|---|
| Xuzhou Municipal Science and Technology Bureau | KC23306 | Dehui Yin |
| Jiangsu Commission of Health | Z2023080 | Dehui Yin |
| Qinglan Project of Jiangsu Province of China | 2024 | Dehui Yin |

## AUTHOR CONTRIBUTIONS

Liping Guo, Data curation, Methodology | Qichuan Pei, Data curation, Methodology | Shiqi Zhao, Data curation, Methodology | Xinru Qi, Data curation | Yixiao Chen, Data curation | Peifei Yang, Data curation | Yangguang Du, Data curation | Mingjun Sun, Writing – review and editing | Dehui Yin, Conceptualization, Funding acquisition, Writing – original draft | Tiansong Zhan, Conceptualization, Writing – review and editing

## DATA AVAILABILITY

The data sets supporting the conclusions of this article are included within the article and its supplemental material.

## ETHICS APPROVAL

All procedures involving animals or animal samples adhered to ethical guidelines and obtained approval from the Animal Care and Ethics Committee of Xuzhou Medical University (ethical approval no. 201801W005) and Animal Research: Reporting of In Vivo Experiments guidelines. The study complies with local and national guidelines and regulations.

## ADDITIONAL FILES

The following material is available online.

### Supplemental Material

**Data S1 (Spectrum00516-25-s0001.xlsx).** Epitope table export results from IEDB of fusion protein 2.
**Data S2 (Spectrum00516-25-s0002.xlsx).** Raw data for I-ELISA.
**Supplemental figures (Spectrum00516-25-s0003.docx).** Fig. S1 and S2.
**Supplemental material (Spectrum00516-25-s0004.docx).** Protocols for endotoxin removal and endotoxin detection; Table S1.

### Open Peer Review

**PEER REVIEW HISTORY (review-history.pdf).** An accounting of the reviewer comments and feedback.

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
