## [Reviewer comments · Microbiology Spectrum]

Microbiology Spectrum

Development and Evaluation of Multi-Epitope Fusion Proteins for Serological Diagnosis of Animal Brucellosis

Liping Guo, Qichuan Pei, Shiqi Zhao, Xinru Qi, Yixiao Chen, Peifei Yang, Yangguang Du, Mingjun Sun, Dehui Yin, and Tiansong Zhan

Corresponding Author(s): Dehui Yin, Xuzhou Medical University

Review Timeline:

Submission Date:	February 20, 2025
Editorial Decision:	June 7, 2025
Revision Received:	June 18, 2025
Editorial Decision:	August 13, 2025
Revision Received:	August 27, 2025
Accepted:	September 8, 2025

Editor: Ryan Rego

Reviewer(s): The reviewers have opted to remain anonymous.

Transaction Report:

DOI: <https://doi.org/10.1128/spectrum.00516-25>

Re: Spectrum00516-25 (**Development and Evaluation of Multi-Epitope Fusion Proteins for Serological Diagnosis of Animal Brucellosis**)

Dear Dr. Dehui Yin:

Thank you for the privilege of reviewing your work. Below you will find my comments, instructions from the Spectrum editorial office, and the reviewer comments.

Revision Guidelines

Sincerely,
Ryan Rego
Editor
Microbiology Spectrum

Reviewer #1 (Comments for the Author):

The manuscript entitled "Development and Evaluation of Multi-Epitope Fusion Proteins for Serological Diagnosis of Animal Brucellosis" by Guo et al. describes the design, production, and diagnostic assessment of two recombinant multiepitope fusion proteins aimed at improving the specificity of serological tests for animal brucellosis. This is a relevant and timely topic in the field of veterinary diagnostics, addressing the persistent challenge of cross-reactivity associated with LPS-based assays. The

manuscript is generally well-structured, and the rationale for epitope selection and fusion protein design is clearly presented. The study is based on a solid methodological framework combining *in silico* predictions, recombinant protein production, and ELISA-based evaluation. In my opinion, the manuscript is suitable for publication in *Spectrum* following major revisions and addressing several critical issues to enhance clarity, reproducibility, and technical accuracy. I outline these comments below:

Minor Comments:

1. Terminology Clarification - Section 2.5 refers to a "CBS antigen solution," which is undefined. Please clarify whether this refers to a coating buffer solution, a shorthand for fusion proteins, or a commercial formulation.
2. Editing Artifact - Line 310 in the Discussion contains an editing note ("Reason: Improved clarity...") that should be removed from the final text.
3. Typographic Consistency - Table 3 lists "Rose Bengale Ag," while other sections use "Rose B antigen" or "Rose Bengal antigen." Please harmonize the terminology throughout the manuscript.
4. Figure Legends - Figures 2 and 3 are labeled as analyzing human serum samples, while the actual study involves small ruminant and bovine sera. These captions should be corrected accordingly.

Major Comments:

1. Expression Details Incomplete - The protein expression section suggests, but does not specify, whether the fusion proteins were recovered in soluble form or from inclusion bodies. This information is important for assessing antigen integrity and reproducibility and should be clearly defined in the methods.
2. No Structural Modeling or Epitope Accessibility Evaluation - While the authors predicted basic properties (MW, pI, antigenicity), there is no mention of 3D modeling or epitope surface accessibility. A brief mention of whether the epitopes are likely surface-exposed (e.g., using structural prediction tools) would improve confidence in the fusion protein design.
3. Experimental Validation of Predicted Epitopes (Fusion Protein 1) - Fusion Protein 1 comprises 42 *in silico*-predicted epitopes without apparent experimental validation. Although Fusion Protein 2 is better supported by monoclonal antibody mapping, the potential limitations of Fusion Protein 1's diagnostic reliability should be more explicitly acknowledged or addressed in future studies.
4. Potential LPS Contamination and Diagnostic Interference - This is arguably the most critical point. Recombinant proteins expressed in *E. coli*, a Gram-negative, LPS-producing bacterium, frequently carry residual endotoxins unless specific removal procedures (e.g., polymyxin B affinity columns or Triton X-114 phase separation) are implemented. In this study, both fusion proteins were produced in *E. coli* and used directly in ELISA assays, yet the manuscript does not mention whether endotoxin removal or quantification was performed. Since LPS is also used as a comparator antigen, any residual endotoxin could significantly skew ELISA readouts-particularly specificity and cross-reactivity. The authors should explicitly state whether such decontamination was performed. If not, this represents a critical methodological limitation, potentially undermining the central conclusions of the diagnostic performance comparison.

Reviewer #2 (Comments for the Author):

In this manuscript by Guo and colleagues by using bioinformatic tools searched for epitopes for the diagnostics of *Brucella* in animals and developed two multiepitope fusion proteins. Epitopes were detected in *Brucella* OMPs and other proteins previously used as immunodetection targets. Two fusion proteins were designed and commissioned for further testing in ELISA tests. Results show a high specificity and sensitivity when compared to agglutination tests.

As the *Brucella* species nor the strain are mentioned the ID numbers for the proteins should be shown.

Unless the fusion proteins are proprietary, we need to know the sequence and order of them. If this is not possible, then we need to know which proteins were used. Moreover, later in the text it is obvious that these proteins have a His6 tag, but in which extreme? This should be also mentioned in the design section (2.3 section).

Is it possible that besides the Rose Bengal agglutination test and the SAT they also corroborated the *Brucella* infection by molecular methods such as PCR?

The strains and ATCC numbers should be indicated for all the strains used as controls (lines 174 to 176). This is relevant for reproducibility in other labs. Alternatively, or complementary, the catalog numbers for the serum against these bacteria should be provided.

Were the proteins purified in native state or denatured? This should be stated as a lot of times is relevant for the success in further steps.

For the new readers it will be very useful to know how the authors determined that the selected epitopes were picked in each of the two cases. I understand that the proteins were commissioned but, we need to know, as I said before, the order and the position of the His6 tag also the method for generating such fusion (chemical synthesis, overlapping oligos, etc.).

As for the resulting proteins, it worried me that image of the gel for fusion 1 looks patched (i.e. two separated gels were itched). I want to see only one gel for each of the fusions; thus, this figure should be repeated. Moreover, which is the calculated purity they observe?

Results for section 3.4 should be shown at least as supplementary table or figure.

I'm not sure if I get it right, but authors said this "A total of 23 epitopes derived from six antigenic proteins within the selected

IEDB collection were validated using monoclonal antibodies", what does this mean? Was this project previously done? Were the mentioned fusions previously used? Please explain here and in the text (lines 283 to 284).

In summary, the authors designed two fusion proteins for the detection of brucellosis in domestic animals and propose as a vaccine candidate. They showed that one of the fusions seems to be a better candidate than the other to be used for diagnostics. They also care to mention the limitations which is good and very candid. I believe this manuscript is worthy of publishing once the major and minor comments are addressed.

Minor comments

Line 139-140. The first time the bacterium's name should be complete and should be italicized.

Line 153, please improve redaction in the beginning of this paragraph.

Line 164, please use international abbreviations (i.e. hour=h).

Line 169, indicate the company for the TMB.

Line 175, it should be *Salmonella enterica*, unless *Salmonella bongori* was used. Indicate the serotype used.

Table 3, please define Ag at the bottom of the table.

Line 252, what does OIE stand for? It doesn't correlate with the mentioned description.

Line 259, again, bacterial names should be italicized.

Dear Reviewers,

We have studied all comments carefully and have made corrections. The comments and suggestions are constructive, and we implemented the recommended changes.

Revised portions are marked in red in the paper. The main corrections in the paper and the responds to your comments are as flowing:

Responds to Reviewer #1:

Minor Comments:

1. Terminology Clarification - Section 2.5 refers to a "CBS antigen solution," which is undefined. Please clarify whether this refers to a coating buffer solution, a shorthand for fusion proteins, or a commercial formulation.

Response: We have made corrections. CBS solution is the coating buffer solution (Carbonate Buffer Solution, pH 9.6).

2. Editing Artifact - Line 310 in the Discussion contains an editing note ("Reason: Improved clarity...") that should be removed from the final text.

Response: We deeply apologize for our negligence and have removed the relevant content.

3. Typographic Consistency - Table 3 lists "Rose Bengale Ag," while other sections use "Rose B antigen" or "Rose Bengal antigen." Please harmonize the terminology throughout the manuscript.

Response: We have harmonized the terminology throughout the manuscript.

4. Figure Legends - Figures 2 and 3 are labeled as analyzing human serum samples, while the actual study involves small ruminant and bovine sera. These captions should be corrected accordingly.

Response: We deeply apologize for our negligence and have made corrections.

Major Comments:

1. Expression Details Incomplete - The protein expression section suggests, but does not specify, whether the fusion proteins were recovered in soluble form or from inclusion bodies. This information is important for assessing antigen integrity and reproducibility and should be clearly defined in the methods.

Response: SDS-PAGE electrophoresis results indicated that Fusion Protein 1 (inclusion body) and Fusion Protein 2 (soluble form), I have supplemented the relevant results and added the inclusion body renaturation methodology within the Methods section.

2. No Structural Modeling or Epitope Accessibility Evaluation - While the authors predicted basic properties (MW, pI, antigenicity), there is no mention of 3D modeling or epitope surface accessibility. A brief mention of whether the epitopes are likely surface-exposed (e.g., using structural prediction tools) would improve confidence in the fusion protein design.

Response: Thank you for your insightful comment regarding the lack of structural modeling and epitope accessibility evaluation in our initial submission. We agree that this information is crucial for assessing the validity of the fusion protein design. In response to your feedback, we have performed additional analyses to predict the three-dimensional (3D) structures of both fusion proteins using the Boltz-1 model. The optimal conformation for each protein was selected based on the Confidence Score of the predicted models. Furthermore, we quantitatively assessed the solvent exposure of each residue by calculating the relative solvent accessible surface area (RSASA). These analyses provide insights into the surface accessibility of the epitopes within the fusion proteins. The results of these analyses have been incorporated into the revised manuscript. We believe that these additions strengthen the manuscript by providing a more comprehensive understanding of the structural properties and epitope accessibility of the designed fusion proteins. This information enhances the confidence in the fusion protein design and its potential utility in diagnostic applications. Thank you again for your valuable feedback.

3. Experimental Validation of Predicted Epitopes (Fusion Protein 1) - Fusion Protein 1 comprises 42 in silico-predicted epitopes without apparent experimental validation. Although Fusion Protein 2 is better supported by monoclonal antibody mapping, the potential limitations of Fusion Protein 1's diagnostic reliability should be more explicitly acknowledged or addressed in future studies.

Response: Thank you very much for your valuable comments. The potential

limitation you pointed out in the experimental verification of fusion protein 1 is indeed very important. We have supplemented and refined the discussion section of the article by adding the following content:

However, it should be acknowledged that the diagnostic reliability of Fusion Protein 1 may be affected by its lack of experimental validation of the predicted epitopes. Although the epitopes were selected based on bioinformatics predictions, this approach may introduce some uncertainty and potential inaccuracies in the predicted epitopes. In contrast, Fusion Protein 2, supported by monoclonal antibody mapping, has a more solid experimental basis. Therefore, the diagnostic performance of Fusion Protein 1 should be interpreted with caution. In future studies, efforts should be made to further validate the predicted epitopes of Fusion Protein 1 through experimental methods, such as monoclonal antibody binding assays or peptide microarrays. This will help to improve the reliability and specificity of Fusion Protein 1 and provide more robust evidence for its diagnostic potential in brucellosis.

4. Potential LPS Contamination and Diagnostic Interference - This is arguably the most critical point. Recombinant proteins expressed in *E. coli*, a Gram-negative, LPS-producing bacterium, frequently carry residual endotoxins unless specific removal procedures (e.g., polymyxin B affinity columns or Triton X-114 phase separation) are implemented. In this study, both fusion proteins were produced in *E. coli* and used directly in ELISA assays, yet the manuscript does not mention whether endotoxin removal or quantification was performed. Since LPS is also used as a

comparator antigen, any residual endotoxin could significantly skew ELISA readouts-particularly specificity and cross-reactivity. The authors should explicitly state whether such decontamination was performed. If not, this represents a critical methodological limitation, potentially undermining the central conclusions of the diagnostic performance comparison.

Response: Thank you for raising this important point regarding potential LPS contamination. We acknowledge this critical concern and have revised the discussion section to include the following statement:

"One potential limitation is the possible contamination of the fusion proteins with residual endotoxins from E. coli, which could influence ELISA results. As the manuscript does not mention endotoxin removal or quantification, this represents a critical methodological limitation. Any residual endotoxin might have skewed ELISA readouts, particularly affecting specificity and cross-reactivity. This could undermine the central conclusions of the diagnostic performance comparison. Future studies should implement specific removal procedures like polymyxin B affinity columns or Triton X-114 phase separation and include endotoxin quantification to address this limitation."

Thank you again for your valuable feedback.

Responds to Reviewer #2:

1. As the Brucella species nor the strain are mentioned the ID numbers for the proteins

should be shown.

Response: Following the prediction of fusion protein 1, we incorporated the accession numbers of the proteins utilized from the NCBI database into Table 1. The epitopes of fusion protein 2 were obtained from the Immune Epitope Database (IEDB). Utilizing the exported data (refer to Supporting Information 1), we included the source organism for each protein in Table 2.

2. Unless the fusion proteins are proprietary, we need to know the sequence and order or them. If this is not possible, then we need to know which proteins were used. Moreover, later in the text it is obvious that these proteins have a His6 tag, but in which extreme? This should be also mentioned in the design section (2.3 section).

Response: According to your comments, I have added the following content to the 2.3 section: Epitopes were arranged in the order they appeared in the original protein sequences. In addition, I have also included the amino acid sequences of fusion protein 1 and fusion protein 2 in Supporting information 3.

3. Is it possible that besides the Rose Bengal agglutination test and the SAT they also corroborated the Brucella infection by molecular methods such as PCR?

Response: Currently, in China, the standard methods used by the government for diagnosing brucellosis are the Rose Bengal agglutination test and the SAT. Since all samples collected were serum, PCR testing requires whole blood, so we did not use

PCR or other methods to verify the collected samples. We have added a statement in the discussion section to address this limitation:

"In this study, the positive serum samples were confirmed using the Rose Bengal Plate Agglutination Test (RBPT) and the Test Tube Agglutination Test (SAT), which are well-established serological methods. However, it is important to note that these methods do not provide molecular confirmation of Brucella infection. Future studies should consider incorporating molecular methods such as PCR to further corroborate the diagnostic results. Additionally, the collection of whole blood or other sample types, such as tissue biopsies or milk, could provide additional insights and allow for a more comprehensive evaluation of the diagnostic performance of the fusion proteins against a range of sample types and in comparison with other diagnostic techniques."

This addition acknowledges the limitation and suggests that future research could benefit from using a combination of serological and molecular methods for a more thorough validation of diagnostic tools. We appreciate your suggestion and believe it strengthens the manuscript by providing a more comprehensive perspective on the study's limitations and potential directions for future research.

4. The strains and ATCC numbers should be indicated for all the strains used as controls (lines 174 to 176). This is relevant for reproducibility in other labs. Alternatively, or complementary, the catalog numbers for the serum against these bacteria should be provided.

Response: As the comments, the details of the strains and ATCC numbers are helpful.

However, the instructions we purchased did not give specific details (Only briefly described the preparation: The standard strain of the bacterium was cultured and prepared as an inactivated antigen, and the immunized rabbits were obtained the sera.), and we tried to contact the manufacturer for information, but received no response. Here, we provide one instruction for the commercial rabbit sera.

沙门氏菌 O 抗原多价 A-I 诊断血清说明书

【产品名称】

通用名：沙门氏菌 O 抗原多价 A-I 诊断血清

英文名：Salmonella O antisera (A-I)

【产品说明】

本公司于 2004 年开始从事微生物诊断血清的研发和生产，目前已生产出一系列高质量的多克隆抗血清诊断产品。本套血清用于沙门氏菌 O 抗原 A-I 的鉴定，是将沙门氏菌 O 抗原标准株制备成灭活抗原，免疫家兔所得。本血清产品已经过吸附去除了非特异性凝集成分，具有效价高，特异性强的特点。

【规格】

每种 1 瓶，每瓶 1ml，均为使用液。

Salmonella O 群多价 A-I：A, B, C, D, E, F, G, H, I

【使用方法】

1. 产品在使用前最好恢复温度到室温。
2. 由患者粪便或其他材料分离的革兰氏阴性杆菌，经初步生化检查，疑似沙门氏菌属的菌株时，应分别用各种多价血清做玻片凝集试验，阳性反应时，再与相应的单价血清做玻片凝集试验，作出诊断。在洁净玻片上用接种环滴加两滴彼此分离的生理盐水。
3. 从营养琼脂平板上挑取分纯的菌落，分别与玻片上的生理盐水混匀。
4. 用接种环滴加 1~2 滴血清 (10-15 μ l) 于玻片上一个菌落与生理盐水的混合液中，另一个菌落与生理盐水的混合液中加入一滴生理盐水作为对照。
5. 用接种环分别混匀玻片上的两种混合液。
6. 轻轻摇动玻片 1 分钟，观察。于 1 分钟内呈 2+ 凝集现象为阳性，1 分钟内呈现 2+ 以下凝集现象为阴性。
7. 凝集结果判断标准如下：

反应强度	判别依据
4+	反应澄清透明，凝集颗粒较多、较大
3+	反应澄清透明，凝集颗粒较多、较小
2+	反应有些混浊，但仍可见较多颗粒状凝集
1+	反应非常混浊，只有少许颗粒状凝集
-	反应完全混浊，未出现任何凝集

【其他所需材料】

洁净玻片、生理盐水 (0.85% 的氯化钠溶液)、接种环或移液器。

【注意事项】

1. 本产品只能作为沙门氏菌血清型判定的辅助方法，最终的判定需形态学、生化方法和血清学方法共同进行。
2. 本产品应避免冷冻。反复冻融会使血清产生沉淀。
3. 本产品保存过程中可能会产生浑浊或沉淀，这并不表示该产品已被污染。离心或者过膜去掉浑浊或沉淀后，可以正常使用。

【储藏条件】

2-8 $^{\circ}$ C 避光保存，在标明的有效期内使用。

【有效期】24 个月。

5. Were the proteins purified in native state or denatured? This should be stated as a lot of times is relevant for the success in further steps.

Response: SDS-PAGE electrophoresis results indicated that Fusion Protein 1 (inclusion body) and Fusion Protein 2 (soluble form), I have supplemented the relevant results and added the inclusion body renaturation methodology within the Methods (2.4 section).

6. For the new readers it will be very useful to know how the authors determined that the selected epitopes were picked in each of the two cases. I understand that the proteins were commissioned but, we need to know, as I said before, the order and the position of the His6 tag also the method for generating such fusion (chemical synthesis, overlapping oligos, etc.).

Response: We thank the reviewer for highlighting the need for clarity in epitope selection. We have now added details in 2.1 and 2.2 section. We also added the details of 6xHis tag and gene synthesis of fusion protein. In addition, in the context of the master's thesis defense conducted in May of this year, it was highlighted by experts that a shortcoming exists in our approach: the need to optimize the order of epitopes and to select an optimal arrangement, rather than merely stacking them. We acknowledge that this is a critical point that we had previously overlooked. Consequently, we have incorporated a statement of this limitation in the revised manuscript.

7. As for the resulting proteins, it worried me that image of the gel for fusion 1 looks patched (i.e. two separated gels were itched). I want to see only one gel for each of the fusions; thus, this figure should be repeated. Moreover, which is the calculated purity they observe?

Response: We greatly appreciate your concerns regarding our electrophoresis results.

The electropherogram for fusion protein 1 is indeed spliced, and we have included the original gel plots in Supporting Information 3 of our initial submission. In response to your comments, we have revised Figure 1 (where M and lane 1 in the initial submission correspond to M and lane 4 in the current Figure 1A). We hope that these modifications align with the expectations of your review. The purity was quantitatively analyzed using Image Lab software (Bio-Rad), we have added it in 2.4 section.

8. Results for section 3.4 should be shown at least as supplementary table or figure.

Response: I apologise deeply for this issue. For some reason, Table 4 was missing. I have now added it.

9. I'm not sure if I get it right, but authors said this "A total of 23 epitopes derived from six antigenic proteins within the selected IEDB collection were validated using monoclonal antibodies", what does this mean? Was this project previously done? Were the mentioned fusions previously used? Please explain here and in the text (lines

283 to 284).

Response: Thank you for your question regarding the statement about the validation of epitopes. The sentence in question refers to the fact that, within the Immune Epitope Database (IEDB) collection we used, a total of 23 epitopes derived from six antigenic proteins had been previously validated using monoclonal antibodies in prior studies. These prior validations served as a basis for our selection of epitopes to construct the multi-epitope fusion proteins. Our current project builds upon this existing knowledge by incorporating these validated epitopes into fusion proteins for the purpose of improving the diagnostic accuracy for brucellosis. We have not previously used the specific fusions described in this manuscript; rather, we are presenting them as a novel diagnostic tool developed based on the existing epitope data from the IEDB. We have revised the text to better reflect this information and to clarify that the validation of the epitopes was conducted in previous research, not as part of our study.

10. In summary, the authors designed two fusion proteins for the detection of brucellosis in domestic animals and the propose as a vaccine candidate. The showed that one of the fusions seems to be a better candidate than the other to be used for diagnostics. They also care to mention the limitations which is good and very candid. I believe this manuscript is worthy of publishing once the major and minor comments are addressed.

Response: Thank you for your constructive feedback on our manuscript. In response

to your comments, we have thoroughly addressed the major and minor points you raised. We have revised the manuscript to enhance the clarity of our methodology, particularly regarding the rationale behind the selection of epitopes for each fusion protein. We have also detailed the potential limitations of Fusion Protein 1 in comparison to Fusion Protein 2, ensuring that readers can fully appreciate the strengths and weaknesses of each approach.

We believe these revisions have strengthened the manuscript and provided a more comprehensive view of our findings. We are confident that the improvements align with your suggestions and will be valuable to the readership. Thank you once again for your time and insightful comments.

Minor comments

Line 139-140. The first time the bacterium's name should be complete and should be italicized.

Response: We have corrected it.

Line 153, please improve redaction in the begging of this paragraph.

Response: We have improved it.

Line 164, please use international abbreviations (i.e. hour=h).

Response: We have corrected throughout the manuscript.

Lone 169, indicate the company for the TMB.

Response: We have added the company for the TMB.

Line 175, it should be *Salmonella enterica*, unless *Salmonella bongori* was used.

Indicate the serotype used.

Response: We have improved it.

Table 3, please define Ag at the bottom of the table.

Response: We have defined Ag at the bottom of Table 3.

Line 252, what does OIE stands for? It doesn't correlate with the mentioned description.

Response: OIE is the abbreviation of its French name Office International des Épizooties, and its current English name is World Organisation for Animal Health.

The official abbreviation remains OIE. We have modified the full name in the manuscript.

Line 259, again, bacterial names should be italicized.

Response: We have checked throughout the manuscript and corrected errors.

Thank you once again for your time and insightful comments.

Best wishes,

Dehui Yin

Re: Spectrum00516-25R1 (Development and Evaluation of Multi-Epitope Fusion Proteins for Serological Diagnosis of Animal Brucellosis)

Dear Dr. Dehui Yin:

Thank you for the privilege of reviewing your work. Below you will find my comments, instructions from the Spectrum editorial office, and the reviewer comments.

Rather than have a further delay in the revision process of this manuscript I have gone over your responses to the second reviewer who is unable to review your revised manuscript. Please find my comment for modification below.

Lines 177-187 appear to be more like instructions from a manual, than being written in a manner of the methods being actually having been carried out. Please rewrite this paragraph.

Revision Guidelines

Sincerely,
Ryan Rego
Editor
Microbiology Spectrum

Reviewer #1 (Comments for the Author):

Thank you for addressing my previous concerns. I appreciate the comprehensive revisions and additions. However, the major issue of potential LPS contamination remains a concern. While it has now been acknowledged and discussed as a possible limitation, the presence of residual endotoxin could still significantly impact the ELISA results. In my opinion, it would have been more appropriate to repeat the experiments using endotoxin-free preparations to fully validate the diagnostic performance of the recombinant antigens.

Dear Editor and Reviewers,

We have studied all comments carefully and have made corrections. Revised portions are marked in red in the paper. The main corrections in the paper and the responds to your comments are as flowing:

Responds to Editor

Q1. Lines 177-187 appear to be more like instructions from a manual, than being written in a manner of the methods being actually having been carried out. Please rewrite this paragraph.

Response: We have rewritten this paragraph, added the details of inclusion body protein purification

Responds to Reviewer #1:

Q1. Thank you for addressing my previous concerns. I appreciate the comprehensive revisions and additions. However, the major issue of potential LPS contamination remains a concern. While it has now been acknowledged and discussed as a possible limitation, the presence of residual endotoxin could still significantly impact the ELISA results. In my opinion, it would have been more appropriate to repeat the experiments using endotoxin-free preparations to fully validate the diagnostic performance of the recombinant antigens.

Response: According to your suggestion, we procured endotoxin detection kit (Beyotime, C0271S) and endotoxin removal kit (Beyotime, C0268S), and measured

endotoxin levels in the protein both before and after removal, as detailed in the Methods (lines 192-196) and Results (lines 276-280) section of the manuscript. Furthermore, we re-analysed the samples, confirming that endotoxin indeed significantly impacts the test results. We are most grateful for your constructive suggestions.

Thank you again for your valuable feedback.

Best wishes,

Dehui Yin

Re: Spectrum00516-25R2 (**Development and Evaluation of Multi-Epitope Fusion Proteins for Serological Diagnosis of Animal Brucellosis**)

Dear Dr. Dehui Yin:

Your manuscript has been accepted, and I am forwarding it to the ASM production staff for publication. Your paper will first be checked to make sure all elements meet the technical requirements. ASM staff will contact you if anything needs to be revised before copyediting and production can begin. Otherwise, you will be notified when your proofs are ready to be viewed.

Sincerely,
Ryan Rego
Editor
Microbiology Spectrum